# Effects of pH and Temperature on Water under Pressurized Conditions in the Extraction of Nutraceuticals from Chaga (*Inonotus obliquus*) Mushroom

**DOI:** 10.3390/antiox10081322

**Published:** 2021-08-23

**Authors:** Ibrahim M. Abu-Reidah, Amber L. Critch, Charles F. Manful, Amanda Rajakaruna, Natalia P. Vidal, Thu H. Pham, Mumtaz Cheema, Raymond Thomas

**Affiliations:** 1School of Science and the Environment, Grenfell Campus, Memorial University of Newfoundland, Corner Brook, NL A2H 5G4, Canada; alcritch@mun.ca (A.L.C.); cfmanful@grenfell.mun.ca (C.F.M.); atrajakaruna@grenfell.mun.ca (A.R.); nprietovidal@grenfell.mun.ca (N.P.V.); tpham@grenfell.mun.ca (T.H.P.); mcheema@grenfell.mun.ca (M.C.); 2The Functional Foods Sensory Laboratory, Grenfell Campus, Memorial University of Newfoundland, Corner Brook, NL A2H 5G4, Canada

**Keywords:** chaga (*Inonotus obliquus*), accelerated solvent extraction (ASE), functional foods, swiss water process (SWP), green extraction, total antioxidant activity, total phenolic content, B-complex vitamins, fat-soluble vitamins

## Abstract

Currently, there is increased interest in finding appropriate food-grade green extraction systems capable of extracting these bioactive compounds from dietary mushrooms for applications in various food, pharmacological, or nutraceutical formulations. Herein, we evaluated a modified Swiss water process (SWP) method using alkaline and acidic pH at low and high temperature under pressurized conditions as a suitable green food grade solvent to obtained extracts enriched with myco-nutrients (dietary phenolics, total antioxidants (TAA), vitamins, and minerals) from Chaga. Ultra-high performance liquid chromatography coupled to high resolution accurate mass tandem mass spectrometry (UHPLC-HRAMS-MS/MS) was used to assess the phenolic compounds and vitamin levels in the extracts, while inductively coupled plasma mass spectrometry (ICP-MS) was used to determine the mineral contents. Over 20 phenolic compounds were quantitatively evaluated in the extracts and the highest total phenolic content (TPC) and total antioxidant activity (TAA) was observed at pH 11.5 at 100 °C. The most abundant phenolic compounds present in Chaga extracts included phenolic acids such as protocatechuic acid 4-glucoside (0.7–1.08 µg/mL), syringic acid (0.62–1.18 µg/mL), and myricetin (0.68–1.3 µg/mL). Vitamins are being reported for the first time in Chaga. Not only, a strong correlation was found for TPC with TAA (r-0.8, <0.0001), but also, with individual phenolics (i.e., Salicylic acid), lipophilic antioxidant activity (LAA), and total antioxidant minerals (TAM). pH 2.5 at 100 °C treatment shows superior effects in extracting the B vitamins whereas pH 2.5 at 60 and 100 °C treatments were outstanding for extraction of total fat-soluble vitamins. Vitamin E content was the highest for the fat-soluble vitamins in the Chaga extract under acidic pH (2.5) and high temp. (100 °C) and ranges between 50 to 175 µg/100 g Chaga. Antioxidant minerals ranged from 85.94 µg/g (pH7 at 100 °C) to 113.86 µg/g DW (pH2.5 at 100 °C). High temperature 100 °C and a pH of 2.5 or 9.5. The treatment of pH 11.5 at 100 °C was the most useful for recovering phenolics and antioxidants from Chaga including several phenolic compounds reported for the first time in Chaga. SWP is being proposed herein for the first time as a novel, green food-grade solvent system for the extraction of myco-nutrients from Chaga and have potential applications as a suitable approach to extract nutrients from other matrices. Chaga extracts enriched with bioactive myconutrients and antioxidants may be suitable for further use or applications in the food and nutraceutical industries.

## 1. Introduction

Functional foods refer to food products or ingredients that are useful in providing known health benefits beyond basic nutritional requirements [1]. Mushrooms are well recognized as high quality sources of functional ingredients with respect to having enhanced levels of vitamins, proteins, and fiber [2]. Historically, mushrooms have rich traditions in folk medicine in many cultures across the globe, as well as popularity as food ingredient in many different cuisines. For example, the well characterized sporocarp are best known for their medical treatments or therapies in many Asian countries [3]. Recent studies have shown the health-promoting properties of mushroom to be associated with their ability to scavenge reactive oxygen species, decrease inflammation, protect against tumors, provide insulin resistance in type 2 diabetes, and bolster the body’s immune system [3,4]. One of the most common dietary mushrooms used as a functional food ingredient and having several health benefits in traditional and folk medicine is Chaga (*Inonotus obliquus*) [5]. Chaga mushroom is a member of the Hymenochaetaceae family [6] commonly found on birch trees within the colder (boreal) climates of North America, Northern Europe, and Asia [7].

Morphologically, *I. obliquus* first develops a white heart rot in the host tree. The chaga spores enters trees or the poorly healed branch stubs through their wounds. During the infection cycle, penetration of the tree wood takes place only around the sterile exterior mycelial mass [8]. The chaga fungus causes decays within the living tree for up to several decades. In the living trees, only the sterile mycelia masses (the black exterior conks) are produced. *I. obliquus* will begin to produce fertile sporocarp underneath the bark. These masses emerge as a whitey mass that turn to brown with time. The fruiting body is rarely observed, as the sexual stage comes almost about fully beneath the bark. These fruiting bodies produce basidiospores which can spread the infection to other susceptible trees.

*Inonotus obliquus* commonly found on birch trees within the colder (boreal) climates of North America, Northern Europe, and Asia [7].

As such Chaga distribution is reflective of the circumboreal forest ecosystem (Figure 1) in the Holarctic regions of Canada, the USA, Asia, and Europe (mainly Russia, Japan, Siberia, and Korea). In these regions, Chaga is highly valued for its use as a therapeutic drink and in enhancing the immune system in traditional or folk medicine [8]. The health benefits ascribed to enhancing the immune system functions are associated with beta-glucans activities. Furthermore, Chaga is reported to suppress the growth of cancer cells via decreased expression of the beta-catenin pathway, regulation of gut bacteria, and its high antioxidant activity [5,9,10]. Previous studies have also reported that Chaga polyphenols and superoxide dismutase enzyme form a potent antioxidant system associated with reduced oxidative stress, diabetes, cancer, and cardiovascular disease risks [11,12].

These health benefits of Chaga are also due to its high myco-nutrient content [13]. Chaga is reported to be a rich source of macrominerals such as potassium (K), nitrogen (N), calcium (C), and a reasonable source of magnesium (Mg), chlorine (Cl), phosphorus (P), sodium (Na), rubidium (Rb), sulfur (S), and manganese (Mn) [14], as well as trace or antioxidant minerals such as iron (Fe), copper (Cu), zinc (Zn), selenium (Se), and the micronutrient iodine (I). Regarding vitamins, Chaga contains B vitamins such as vitamin B1 (Thiamine), vitamin B2 (Riboflavin), vitamin B3 (Niacin), vitamin D2 (ergocalciferol), vitamins A, and vitamin K [15]. Besides the abundance of these vitamins, Chaga also contains significant levels of dietary phenolics which also act as antioxidants [16]. Considering the rich myconutrient composition of Chaga, there is great interest in the food and nutraceutical industries to find suitable food-grade, green extraction techniques that will permit the recovery of high contents of these mycochemicals from Chaga for applications in functional foods, nutraceuticals, and pharmacological formulations or innovations.

Many different methods have been used historically to extract myco-chemicals from a variety of natural sources [17,18]. To this end, organic solvents such as ethanol, methanol, acetone, ethyl acetate—or their mixture with water, dichloromethane, and hexane—either at ambient temperature or at the boiling point of the chosen solvent(s) are often used. Longer treatment times, higher temperatures, large quantities of solvent required, decomposition of thermo-labile myco-chemicals, environmental health and safety risks are challenges associated with some of the organic solvents typically employed for extracting bioactive compounds from Chaga [19].

Today, innovative extraction methods are explored continuously to overcome these challenges. The Swiss water extraction method is an example of a green system that is considered food-grade, and represents an environmentally friendly technique combining water, temperature, osmosis, pressure, and extraction time to remove primarily caffeine from coffee beans [20]. This method was designed to reduce energy utilization, facilitate the use of alternative water-based solvents geared at producing high-quality safe extracts and isolates from a range of matrices [19]. Currently, there is a paucity of information in the scientific literature concerning how pH modulates solvents’ extraction efficiency using the Swiss water extraction method under pressurized conditions and even less is known when applied to extracting myco-chemicals from Chaga matrices. Recently, boiling water extraction was demonstrated to be successful in the extraction of bioactive mycochemicals from Chaga sourced from different geographical areas for various medicinal applications in oncology [21]. Boiling water extraction method represents the most common approach to obtain Chaga extracts, and tea in many cultures and laboratory applications currently. We hypothesize that pH will enhance the extraction efficiency of water-based extraction solvents under pressurized conditions and could be useful in extracting high levels of functional ingredients from Chaga mushrooms.

Herein, we proposed to use a modified version of the Swiss water extraction method under pressurized conditions at varying pH as a green solvent or potential food-grade approach suitable for recovering high levels of myconutrients from Chaga.

## 2. Materials and Methods

### 2.1. Chemicals

Standards of water-soluble vitamins including thiamine hydrochloride (vitamin B1), riboflavin (vitamin B2), nicotinamide (vitamin B3), pantothenic acid (vitamin B5), pyridoxine (vitamin B6), folic acid (vitamin B9), and cobalamin (vitamin B12) were purchased from Sigma Aldrich (St. Louis, USA). Likewise, polyphenol standards including quercetin, trans-ferulic acid, ellagic acid, gallic acid, ρ-hydroxybenzoic acid, salicylic acid, protocatechuic acid, caffeic acid, and vanillic acid were acquired from Sigma Aldrich (St. Louis, MO, USA). ICP-MS 43-element standard mix (IV-ICPMS-71A) was purchased from Inorganic Ventures (Christiansburg, Virginia, USA). Tris(2-carboxyethyl)phosphine hydrochloride (TCEP), xylenol orange, ferrous ammonium sulfate hexahydrate, ortho-dianisidine dihydrochloride, meta-phosphoric acid, glacial acetic acid (HPLC grade), methanol (HPLC grade), acetone (HPLC grade), acetonitrile (HPLC grade), 2,4,6-tripyridyl-s-triazine (TPTZ), ferric chloride hexahydrate, anhydrous sodium sulfate, ethanol (95% *v/v*), hydrogen peroxide (30%, *v/v*), sulfuric acid (98% *v/v*), sodium phosphate, hydrochloric acid (36% *v/v*), ortho-phosphoric acid (85% *v/v*), methylchroman-2-carboxylic acid (Trolox), sodium hydroxide, glycerol, sodium carbonate, and Folin–Ciocalteu reagent were purchased from Sigma Aldrich (Oakville, ON, Canada). Nitric Acid (TraceMetal™ Grade) and Dionex ASETM Prep DE diatomaceous earth were obtained from Thermo-Fisher Scientific (Ottawa, ON, Canada).

### 2.2. Preparation of Chaga Sporocarp Extracts

The samples of Chaga mushroom were collected from the province of Newfoundland, Canada. After collection Chaga was powdered using a mechanical mill (Hamilton Beach, purchased from Amazon.ca). Then 5 g samples were filled into 10 mL stainless-steel accelerated solvent extractor (ASE) cell sample holders and extracted using a Dionex 350 ASE (Thermo Scientific, Waltham, MA, USA). Ionized water at different pH values (pH = 2.5, 7.0, 9.5, and 11.5) was used as extraction solvents (SD501, Enagic, Toronto, Canada). The ASE extractions were conducted based on the following program: (number of cycles = 2; pressure = 1500 psi; static time = 5 min; flush ratio = 50% rinse, and temperature = 60 °C and 100 °C). The extracts recovered were lyophilized using a freeze dryer (FreeZone freeze dry system, Labconco, Kansas City, MO, USA) and the dried extracts stored in amber-colored vials at −20 °C until further chemical analysis.

### 2.3. Total Antioxidant Activity (TAA) Analysis

The TAA of the mushroom extracts was measured colorimetrically using the ferric reducing antioxidant power (FRAP) method [22], with minor modifications. Briefly, 20 µL of each standard or sample was reacted with 180 µL of freshly made FRAP solution, and the resultant mixture incubated in the dark for 120 min. Absorbance measurements were recorded at 593 nm on a Cytation imaging microplate reader (BioTek, Winooski, VT, USA). The TAA extracts’ values were determined by summing the lipophilic (LAA) and hydrophilic (HAA) antioxidant activity values based on a Trolox standard curve using the following concentrations (0.5, 1, 1.5, 2, 2.5, 3, 3.5 mM). The results for the Chaga extracts were expressed in µmol Trolox equivalent (TE)/g extract (dry weight: DW).

### 2.4. Total Phenolic Content (TPC) Analysis

The TPC of mushroom extracts was measured using the Folin-Ciocalteu reagent as described previously [23], with minor modifications. Thus, 25 µL of each standard or mushroom extract was mixed with 125 µL of a 10-fold diluted Folin–Ciocalteu reagent in microplate wells. Acidified ethanol (50 µL, 0.7% *v/v*) or sodium phosphate buffer (50 µL, 50 mM, pH 7.5) was added to analyze the lipophilic or hydrophilic phenolic content in the extracts, respectively. The resultant mixtures were incubated in darkness for 120 min, and the absorbances measured at 755 nm on a Cytation imaging microplate reader (BioTek, Winooski, VT, USA). The TPC of the mushroom extracts were calculated by summation of the hydrophilic phenolic content (HPC) and lipophilic phenolic content (LPC) and the concentration measured based on quercetin standard curve. The results were expressed as µmol quercetin equivalent/g extract (DW).

### 2.5. B-Vitamin (B1, B2, B3, B5, B6, B9, and B12) Analysis by Ultra-High-Pressure Liquid Chromatography-Mass Spectrometry (UHPLC-MS)

The determination of B vitamins (water-soluble) in freeze-dried mushroom extracts was based on the method of Akhavan et al. [24]. Briefly, a stock solution of vitamin B standard mix (B2, B3, B5, B6, and B9) at a concentration of (1 mg/mL) was diluted serially with distilled water to obtain B vitamin calibration standards in the range of 5–100 µg/mL Aliquots of the extracts (1 mg) were diluted with distilled water (10 mL) and spiked with hippuric acid as an internal standard. The resultant mixture was homogenized, then incubated in darkness for 45 min at 4 °C, followed by centrifugation at 25,200× *g* for 15 min at 0 °C. The supernatant was collected, pooled, and filtered into amber-colored vials for analysis by UHPLC-HRAMS-MS/MS. For the analysis of Vitamin B, a Luna C18 column (100 × 2.0 mm I.D., particle size: 3 µm, pore diameter: 10 nm) purchased from Phenomenex (Torrance, CA) was employed. The solvent system used was as follows: solvent A consisted of 10 mM of ammonium formate and 0.1% Formic Acid, while solvent B consisted of 100% acetonitrile. Chromatographic separation was performed at 35 °C (column oven temperature) with a flow rate of 0.3 mL/min, and 10 µL of the extract was injected into the system. The following solvent gradient was used for separating the vitamins: ramp at 20 min with 55% solvent B; next transition to 22 min. 0% Solvent B and 100% of Solvent A. The Orbitrap ESI source was run in positive ion mode for data acquisition. The following optimized parameters were used for the Orbitrap mass spectrometer: sheath gas flow rate of 51; an auxiliary gas flow rate: 0; ion spray voltage: 3.20 kV; capillary temperature: 300 °C; S-lens RF: 100 V; capillary voltage: 35 V; mass range: 50–500 *m/z*; full scan mode at a resolution of 60,000 *m/z*; top-3 data-dependent MS/MS at a resolution of 30,000 *m/z*; and collision energy of 35 (arbitrary unit); injection time 22 min; isolation window: 2.0 *m/z*; automatic gain control target: 0.250; with dynamic exclusion setting of 30.0 s. The mass spectrometer was externally calibrated to 1 µg/mL using ESI negative and positive calibration solutions (Thermo Scientific, Berkeley, MO, USA) before usage.

### 2.6. Fat-Soluble Vitamins (A, E, D3, K1, K2, E) Analysis by UHPLC-HRAM-MS/MS

The determination of fat-soluble vitamins present in the freeze-dried mushroom extracts was based on the method of Ramadan and coworkers with minor modifications [25]. Briefly, mushroom extracts were homogenized in 1 mL of methanol: chloroform (1:1) containing hippuric acid as an internal standard. The resultant mixture was incubated for 30 min in darkness at 4 °C followed by sonication for an additional 30 min at 4 °C in a sonication water bath. Thereafter, the samples were homogenized, centrifuged (25,200× *g* for 15 min at 0 °C), then the supernatant was collected into amber-colored vials for LC-MS analysis. A polar acclaim C18 column (150 × 4.6 mm I.D., particle size: 5 µm, pore diameter: 12 nm; Thermo Fisher Scientific, Mississauga, ON, Canada) was used for the resolution of the fat-soluble vitamins. The solvent system used consisted of 5 mM ammonium acetate. The separation was carried out at 40 °C (column compartment temperature) with a flow rate of 0.3 mL/min, and 10 µL of the extract was injected into the machine and elution done isocratically for 32 min. The Orbitrap MS was operated in the positive ESI ion mode during data acquisition. The following optimized parameters were used for the Orbitrap MS: sheath gas flow rate: 25; ion spray voltage: 500 kV; an auxiliary gas flow rate: 2; S-lens RF: 80.00 V; mass range: 60–900 *m/z*; capillary temperature: 320 °C; capillary voltage: 10.0 V; full scan mode at a resolution of 60,000 *m/z*; top-three data-dependent MS/MS at a resolution of 30,000 *m/z*; and collision energy of 35 (arbitrary unit); injection time of 15 min; isolation window: 2.0 *m/z*; automatic gain control target: 0.250; with dynamic exclusion setting of 30.0 s. The mass spectrometer system was externally calibrated to 1 µg/mL using ESI positive and negative calibration solutions (Thermo Scientific, Berkeley, MO, USA) before usage.

### 2.7. Phenolic Compounds Analysis by UHPLC-HRAM-MS/MS

Aliquots (1 mL) of 10% Formic acid: water (*v/v*) was added to 1 mg freeze-dried extracts in centrifuge tubes. The resulting mixture was homogenized, incubated in darkness for 15 min and then centrifuged at 25,200× *g* for 15 min at 0 °C. The supernatant was pooled into amber-colored vials for UHPLC-HRAM-MS/MS analysis. A standard mix of select phenolics were used to create a standard curve covering the following concentration range: 5–100 µg/mL. The UHPLC-HRAMS/MS analysis was conducted on an LTQ Orbitrap XL mass spectrometer (Thermo Scientific, Berkeley, MO, USA) with an automated Dionex UltiMate 3000 UHPLC system controlled by the Chromeleon software v. 7.3. Resolution of the phenolics present in the sample was performed on a Luna C18 column (100 × 2.0 mm I.D., particle size: 3 µm, pore diameter: 10 nm) purchased from Phenomenex (Torrance, CA, USA). The solvent system used was as follows: Solvent A consisted of H_2_O (containing 0.1%, *v/v* Formic acid), and Solvent B consisted of acetonitrile (consisting of 0.1%, Formic acid). Chromatographic separation was done at 25 °C (column temperature) with a flow rate of 0.3 mL/min, and 5 µL of the extract was injected into the machine. The following solvent gradient was used for separating the polyphenols: 0–2 min with 0–10% solvent B; 2–15 min applying 10–45% solvent B; 15–30 min using 45–90% solvent B; 30–35 min using 90–10% solvent B. The Orbitrap MS was operated in the negative ESI ion mode. The optimized parameters used for the Orbitrap mass spectrometer were: sheath gas flow rate: 25; ion spray voltage: 5.0 kV; an auxiliary gas flow rate: 2; capillary temperature: 320 °C; capillary voltage: 10 V; S-lens RF:80 V; mass range:50–1500 *m/z*; capillary voltage: 10 V; full scan mode of 60,000 *m/z* resolution; top-6 data-dependent MS/MS acquisition at a resolution of 30,000 *m/z* and collision energy of 35 (arbitrary unit); isolation window: 2.0 *m/z*, automatic gain control target: 2 e5 with dynamic exclusion setting of 30.0 s and injection time of 15 min. The MS was externally calibrated to 1 µg/mL using ESI positive and negative calibration solutions (Thermo Scientific, Berkeley, MO, USA) before usage.

### 2.8. Extraction and Analysis of Minerals by Inductively Coupled Plasma-Mass Spectrometry (ICP-MS)

Aliquots (1 mg) of freeze-dried Chaga mushroom extracts were digested in 10 mL of trace nitric acid (70% *v/v*) using a Multiwave Go-microwave digestion system (Anton Paar GmbH, Graz, Austria). The microwave digester was operated under the following conditions: temperature ramped to 180 °C for 10 min, then held at this temperature (180 °C) constantly for an additional 20 min. A calibration curve (ranging from 1 to 500 µL^−1^) was prepared using IV-ICPMS-71A standard mixture (InorganicTM Ventures, Inc.; Christiansburg, VA, USA). Aliquots of the digestates were diluted with 2% nitric acid and spiked with rhodium-103 (final concentration = 10 µL^−1^) as the internal standard. The ICP-MS analysis was conducted using the following parameters: auxiliary gas flow of 0.79 L/min, nebulizer gas flow of 1.01 L/min, plasma gas flow of 14 L/min, RF power of 1548 W, detector mode KED, dwell time of 0.01 s using argon gas at 99.99% purity.

### 2.9. Statistical Analysis

ANOVA was used to assess the effects of extraction pH and temperature on the antioxidant activity, phenolic compounds, vitamins, and mineral content in the mushroom extracts. When the means were significantly different between pH and temperature combinations, the means were compared with Fisher’s least significant difference (LSD), α = 0.05. Pearson’s correlation coefficients were used to determine the linear association between subclasses of myco-chemicals (phenolics, vitamins, and minerals) and their contributions to the antioxidant activity of the extracts. Outliers were detected using the Grubbs test. Principal component analysis was used to discern similarities between the extraction conditions (pH and temperatures) and the myco-nutrients recovered. Statistical analyses were performed with XLSTAT (Addinsoft, Paris, France). Figures were prepared using XLSTAT and SigmaPlot 13.0 software programs (Systat Software Inc., San Jose, CA, USA).

## 3. Results and Discussion

The current interest in medicinal mushrooms for their unique and promising health promotive benefits such as anti-inflammatory, anticancer, anti-diabetic, and antioxidant properties is remarkable [26]. Furthermore, consumers have high interests in natural sources of dietary antioxidants for use in their daily food intake. Herein, we demonstrated a modified Swiss water method consisting of elevated temperature and pH under pressurized condition using accelerated solvent extraction as a suitable food grade solvent system to extract myco-chemicals from Chaga mushroom. Chaga mushroom is considered an enriched source of natural antioxidants.

### 3.1. Total Polyphenols Content (TPC) and Total Antioxidant Activity (TAA)

Principal component analysis demonstrated the TPC and TAA clustered together with the extracts obtained at pH 11.5 at 100 °C. The TPC and TAA were also grouped with the HPC, and HAA under these extraction conditions, while the LPC clustered with extracts obtained at pH 2.5/60 °C, pH 2.5/100 °C, and pH 7/100 °C. This grouping captured 96.22% of the total variability in the data set, with PC1 explaining 81.45%, and PC2 accounting for 14.77% of variability in data. TPC and HPC were grouped with Ph 11.5/100 °C treatment, implying that they are best extracted using high pH and temperature from Chaga matrices under pressurized conditions. Furthermore, this pH and temperature combination appears to extract the hydrophilic portion comprising the highly polar phenolic compounds, and this may be noted from the ANOVA analysis, which demonstrates HPC as the major contributor to the TPC. Interestingly, the composition of the solvent system is water based and was not anticipated to extract any significant component of lipophilic polyphenols or antioxidants. However, we observed that although most of the polyphenols and antioxidants extracted were hydrophilic in nature, the lower solvent pH (2.5 and 7) was effective in extracting the lipophilic phenolics from the Chaga matrix, while higher pH (11.5) at 100 °C was effective in extracting LAA from Chaga under pressurized conditions using water as an accelerated solvent (Figure 2). We observed the TAA varied significantly with treatment and ranged from 98.40 to 381.30 μmol TE/g DW, with the highest TAA present in the extract was acquired at pH 11.5 and a temperature of 100 °C. Similarly, the TPC ranged between 102.27 ± 1.39 to 273.06 ± 2.19 μM QE/g Chaga DW, with the highest TPC also observed in the extracts acquired at pH 11.5 and a temperature of 100 °C (Figure 2; Appendix A). These results demonstrate extraction with pH 11.5 at 100 °C has unique conditions and extraction potential towards the acquisition of hydrophilic and lipophilic antioxidants in Chaga mushroom extract using water as an accelerated solvent under pressurized conditions. Hwang and coworkers (2019) [27] reported that phenolic compounds extracted increased in mushrooms treated at 100 °C. Seo and Lee (2010) [28] also noted that phenolic components in Chaga extracts produced by supercritical water increased with temperature, implying that temperature appears to be useful in phenolics extraction from Chaga mushroom. It is worth mentioning that the use of pH, temperature, and pressure can change the physiochemical nature of water to extract organic components that are generally not soluble in aqueous solvents, which is clearly shown from the extraction of the lipophilic phenolic components in our study. The values we observed for TPC and TAA in Chaga extracts are consistent with values observed in the literature [27]. However, huge variation is noted in the literature for the phenolics content, and antioxidant activities measured in Chaga. The variation in antioxidant activities and phenolics content reported in the literature are reported to be influenced by source geography, temperature, carrier birch type, time of harvest, and part used for extraction [29].

The correlational analysis between TAA and TPC showed that a strong antioxidant activity came from the HAA (r = 0.76, *p* < 0.0001) (Figure 3). Interestingly, relatively considerable levels of both LAA and LPC were observed in the extract, considering that an aqueous-based solvent system was used (Figure 2). The elevated temperature and pH used for the extraction under pressurized conditions appear to contribute to the extraction of lipophilic antioxidants and phenolics in this extraction system. Finally, the correlation coefficient (r = 0.78) between TAA and TPC, suggested that approximately 78% of the extracts’ total antioxidants may be attributed to the TPC present in the extract, which was driven by the high association of the HPC with the TAA (r = 0.81). These findings suggest that the hydrophilic phenolics are a major contributor to the TAA of the extracts obtained in this extraction system.

### 3.2. Chaga Extracts Composition

Mushrooms have a long history in folk medicine and ethnopharmacological uses for their health promotive benefits. The rich myco-nutritional composition is attributed to these health promotive benefits [30]. Evaluation of the extracts obtained in this study demonstrated Chaga is a good source of vitamins, dietary polyphenols, antioxidant, trace, and macro minerals.

#### 3.2.1. Minerals

The minerals recovered in the Chaga extracts were grouped as antioxidant minerals, trace, macro, and heavy metals (Table 1) according to their dietary or nutritional relevance. The results suggest significant quantities of antioxidant minerals (Mn, Cu, Zn, and Se), macronutrients (Na, Mg, P, S, K, and Ca), trace minerals (B, Cr, and Co) and to a lesser extent toxic heavy metal (As) were present in the extracts (Table 1). However, the heavy metals extracted are significantly below the tolerable limits recommended for safe human consumption [31] (Institute of Medicine (US) Panel on Micronutrients, 2001). Principal component analysis was used to discern the extraction conditions’ effectiveness in recovering the different classes of minerals from Chaga matrices using the modified Swiss method (SWP). In the PCA biplot, the total microminerals and trace minerals were grouped with pH 2.5 and pH 9.5 at the higher temperature treatments (100 °C). These treatments seem to promote the extraction of macronutrients and trace minerals. The total antioxidant minerals were grouped with a pH of 11.5 and high temperature (100 °C). This may indicate that antioxidant minerals can be better extracted by high temperature and pH from Chaga using the SWP. Furthermore, pH 2.5 at 60 °C was better for extracting heavy metals like Arsenic. Zn was the predominant antioxidant mineral present in the extract with values ranging from 67.37 ± 1.29 to 105.87 ± 3.05 µg/g DW of Chaga, with the highest concentration observed in the extracts collected at pH 2.5 and 9.5 at a temperature of 100 °C (Table 1). Mn, Fe, Cu, and Se were the other antioxidant minerals present in the Chaga extracts. A pH of 2.5 at low temperature (60 °C) was effective for the extraction of Mn from Chaga (9.85 ± 3.05 µg/g DW). This was several folds higher than the values observed at the other pH and temperature combinations (0.0 to 0.539.85 ± 3.05 µg/g DW). Conversely, a pH of 9.5 at temperatures of 60 °C and 100 °C, as well as a pH of 2.5 at a temperature of 60 °C was the most effective in the recovery of Se from Chaga matrices. This indicate pH is a very important determinant in modulating the level of selenium extracted from Chaga using the SWP. We next assessed the contributions of the antioxidant minerals to the total antioxidant activity measured in the extract using Pearson’s correlation. The total antioxidant minerals were moderately correlated (r = 0.52) with the TAA in the extracts recovered (Figure 3 and Appendix A). The results of the mineral contents expressed in µg/g DW of Chaga mushroom is presented in Table 1 along with the recommended daily allowances (RDA’s) [31]. The antioxidant minerals form a very important part of the total minerals found in Chaga. These minerals are often required in low quantities in human nutrition to meet the RDAs [32] and typically supplemented in many food formulations. The recommended dietary allowance (RDA) for Cu, Mn, and Zn has been established. An infusion of Chaga mushroom made from 10 g dry weight (DW) (teaspoon full) could meet the demand for Cu, Mn, Zn and Se which is set at 5%, 10%, 10%, and 4%, respectively [31].

Arsenic (As) was the only heavy metal observed in the extract and the values ranged between 0.81 ± 0.04 to 1.63 ± 0.05 with the highest-level present at pH 2.5 and a temperature of 60 °C. It is important to note the level of As detected in the extract is low. As is lethal in humans in the range of 2–10 mg/kg/day for an average size human [33]. The values reported in these extracts are below the safe levels reported for human consumption. The U.S. Food and Drug Administration (FDA) has established tolerance levels for As to be permissible at levels ranging from 0.5 to 2 µg/mL in different foods [33].

Both the macro and trace minerals clustered with pH 2.5 and a temperature of 100 °C in the PCA biplot suggesting this condition was optimal for the extraction of these minerals from Chaga using the SWP. Ca, Na, and Mg were the predominant macro minerals present in the extracts and their levels were significantly impacted by an extraction pH of 2.5 and 9.5 at 100 °C, where the highest levels for these minerals were observed. The other macrominerals Ca and P recovery were highest at a pH of 2.5 and temperature 60 °C. We observed Ca, Na, and Mg were the predominant macrominerals present in the extracts with significantly higher levels recovered (Table 1). Similarly, boron the predominant trace minerals, was more successfully recovered when extracted at pH 2.5 and a temperature of 60 °C. These results clearly demonstrate that a pH of 2.5 or 9.5 at 100 °C are useful for extracting macro and trace minerals from Chaga using the SWP.

#### 3.2.2. Vitamins (Water- and Fat-Soluble)

Water is an interesting and practical solvent in extraction, and its physiochemical properties can change with varying temperatures. These conditions allow water at near-critical conditions to dissolve hydrophobic compounds [34]. Furthermore, an increased movement/rotation of water molecules can also be observed at higher temperature which allows for dissolving less polar compounds, analogous to that of methanol and ethanol (slightly polar solvents) which in turn may enable the extraction of nonpolar components [34,35]. In this context, we observed that when water was used as a solvent (pressurized fluid) in the SWP evaluated in this study both fat- and water-soluble vitamins were successfully extracted from Chaga matrices. Vitamin E and K were the predominant vitamins observed in the extracts with values with the average of 112, 460 and 2423 ng/g DW Chaga, respectively. The high alkaline and high-temperature treatment (pH 11.5/100 °C) was successful in extracting both classes of vitamins including vitamins D, and K (fat-soluble vitamins), as well as B-vitamins (water-soluble vitamins). On the other hand, total vitamin E (fat-soluble) including Beta-tocopherol analogue was found to be best extracted at the treatment with lower pH (2.5) and higher temperature (100 °C). These findings suggest that high temperature strongly impacts the extraction of the fat and water-soluble vitamins.

From the PCA of fat-soluble vitamins (FSV) and water-soluble vitamins (B-vitamins), it was observed that vitamin K2, total vitamin D, total B-vitamins, and total vit. K are grouped with higher temperature and alkaline pH (9.5 and 11.5) at 100 °C. Alpha-Tocopherol and Vitamin K1 are grouped with pH7 and temperature of 100 °C. On the other hand, vitamin A and beta-tocopherol grouped with pH 2.5/60 °C, pH 2.5/100 °C, and pH 9.5/60 °C extraction conditions.

Despite the large volume of research on Chaga in the scientific literature, to the best of our knowledge there is no work describing the vitamin content in the literature. Vitamin D2 was mentioned and evaluated in different mushroom genotypes and its quantity ranged from 0.3 to 5.3 µg/100 g. However, our results are in the range of 0.02 to 0.04 µg/100 g DW Chaga. Despite the low content of vitamin D in mushrooms, high levels of sterols could be found in mushrooms like ergosterol, as a potential source and precursor of vitamin D2 [36].

The transformation of ergosterol to vitamin D2 can be achieved by exposure to ultraviolet (UV) irradiation [37]. Sunlight exposure is low in the boreal ecosystem compared to other regions, and this, as well as the extraction conditions employed in this study, may account for the low vitamin D levels observed in Chaga extracts.

The effect of solvent pH and temperatures on vitamin A, vitamin D, vitamin K1, vitamin K2, and vitamin E which includes Alpha, tocopherol is shown in Figure 4. Vitamin A content in samples extracted with pH 2.5 and 9.5 at 60 °C are not significantly different from each other. No significant difference was found among the different extraction systems for Vitamin D2 B5.

Vitamin B1 (Thiamin) was reported to be in the range of 0.02–1.6 µg/g DW, while vitamin B3 (Niacin) was found in the range of 1.2–6 µg/g DW in different kinds of mushroom [38]. In Chaga, the best extract contained 0.17 ng/100g of B3 at pH 9.5 and a temperature of 100 °C.

Vitamin B5 content of Chaga samples extracted at pH 2.5, 7, 9.5 at both 60 °C and 100 °C are not significantly different from each other, whereas the samples extracted at pH 11.5, 100 °C is significantly different from samples at pH 2.5 and pH 7 and results in a high content of vitamin B5. Extraction temperature had no significant impact on vitamin B5 content. The lowest vitamin B6 content was found in the mushroom sample extracted at pH 7, 100 °C. The highest and the lowest vitamin B3 content were extracted at pH 9.5, 60 °C, and pH 7, 100 °C, respectively, and these values were significantly different from each other. Vitamin B1 content is highly elevated in samples extracted at pH 9.5, 60 °C and shows the least value at pH 7, 100 °C. Vitamin B2 and Vitamin B12 were undetectable. Vitamin B3 was found in the range from 3.25 to 11.62 ng/100 g in Chaga extracts, with the highest level observed at pH 9.5 and temperature of 100 °C. Although the levels for some of the vitamins recovered using this method are lower than that for vitamins in other mushroom varieties. This work demonstrates for the first time the acquisition of both fat- and water-soluble vitamins simultaneously in Chaga extracts obtained using water as pressurized extraction fluids at varying pH and temperature in the SWP proposed.

#### 3.2.3. Phenolics

The phenolic compounds identified and quantified in the extracts obtained using the SWP is illustrated in Table 2. Analysis of the phenolic composition of the different extractions/treatments resulted in the identification and quantification of twenty-four phenolic compounds in Chaga mushroom extracts following LC-HESI-HRAM-MS/MS analysis. The results for the composition at the best extraction conditions at pH 11.5 and a temperature of 100 °C is presented in Table 2. The identification was based on the molecular formula, retention time, and exact molecular mass (Table 2). The compounds: 3,4-dihydroxybenzaldehyde, 4-methoxycinnamic acid, dihydroxyterephtalic acid, protocatechuic acid, and caffeic acid were found as the highest phenolics in the Chaga extracts produced using the modified Swiss water method (Table 2). It was suggested that phenolics in Chaga are the main components responsible for the antioxidant activity [18], and this is consistent with the high correlations observed between these components in our study.

According to the results obtained, 3,4-dihydroxybenzaldehyde was significantly higher (104.851 µ/mL) when extracted at pH 11.5 and a temperature of 100 °C compared to other treatments. The content found in our research are consistent with previous reports in the literature as this compound was a major phenolic compound observed, but the levels obtained in our work are much higher. Interestingly, 3,4-dihydroxybenzaldehyde has been reported to have significant biological activities such as antioxidative, antibacterial, anti-aging, and anti-inflammatory [39]. The present study also revealed other phenolics like flavonoid aglycones such as myricetin, quercetin, isorhamnetin, and phenolic acids—such as gallic, vanillic, protocatechuic, ferulic, ellagic acid, methyl ellagic, syringic, and salicylic acids—to be major compounds in the Chaga extracts obtained. Total hydroxycinnamic acids (69.6 µg/mL), total hydroxybenzoic acids (129.20 µg/mL), and total phenolics (58.22 µg/mL) were found at higher levels in the extracts obtained using pH 11.5, 100 °C. Phenolic acids like 4-methoxycinnamic_acid (57 µg/mL), 3,4-dihydroxybenzaldehyde (104.85 µg/mL), 3,4-dihydroxybenzalacetone (57.17 µg/mL), and Caffeic acid (11.73 µg/mL) were observed in the extracts. Protocatechuic acid 4-glucoside is the only glycosylated phenolic acid detected along with Syringic acid, which was significantly higher at pH 7 and pH 9.5, and temperature of 60 °C compared with the other treatments. Ferulic acid showed similar trend, though the amounts were lower. Myricetin was significantly (*p* < 0.05) higher in pH 7_60 °C than other pH-temperature treatments. Vanillic acid, ferulic acid, isorhamnetin, and salicylic acid were detected in moderate amounts between 0.5–0.6 µg/mL. It is worth mentioning that the flavonoids (isorhamnetin, myricetin), salicylic acid, and protocatechuic acid 4-glucoside are identified in Chaga mushroom for the first time in our study to the best of our knowledge. These phenolics were detected as aglycones, which might be due to the heat/and or pH impact on the aglyconic linkage with sugars’ moieties. Salicylic acid was significantly (*p* < 0.05) higher at pH 11.5 and temperature of 100 °C compared to the other treatments. One other major phenolic compound detected in Chaga mushroom using the extraction conditions was 3,4-dihydroxybenzaldehyde (104.851 µg/mL). Furthermore, 4-methoxycinnamic acid (57.23 µg/mL) is being reported in Chaga for the first time. This compound was found to have antioxidant, antihyperglycemic, and hypolipidemic activities [40,41,42]. Other new compounds that were identified and quantified in for the first time in Chaga include methyl-ellagic acid and ellagic acid. Both these compounds were found to exist at higher amounts in extracts obtained using pH 11.5–100 °C treatment. These compounds were reported to have anti-radical and strong neuroprotective and antioxidants properties [43]. Quercetin has previously been reported in Chaga and it was also detected in our present work but at lower amounts [27].

#### 3.2.4. Correlations

To get a better understanding of the antioxidant potential of the Chaga extracts obtained as a function of its chemical composition, we assessed the correlations between the TAA and total antioxidant minerals, TPC, vitamins, and dietary phenolics (Figure 5). As previously mentioned, a high correlation between antioxidant capacity and TPC was found, which indicates that phenolic compounds could be the foremost contributors to the antioxidant activity of the Chaga extracts obtained. This is consistent with several reports in the literature demonstrating high association between phenolics and the antioxidant activity [1].

Likewise, the total antioxidant minerals (TAM) were moderately correlated with the TAA (r = 0.52; *p* = 0.016), so too was vitamin E with the LPC (r = 0.64). Vitamin E exists in different forms called ‘Tocopherols’ which are known antioxidants [36]. In our present work, we have determined three types of these antioxidant vitamins, namely, Alpha-, Beta-, and Delta-Tocopherols in the Chaga extracts using the SWP. These results demonstrate that phenolic acids are highly associated with the antioxidant activity present in the Chaga extracts obtained using the modified Swiss water method evaluated in this study. Collectively, pH 11.5, 100 °C appears to be very useful for extracting individual dietary phenolics, TPC, and TAA from Chaga matrices (Figure 3). The correlation results between antioxidant activity, TPC, phenolics, vitamins, and antioxidant minerals suggest the Chaga extracts obtained appears to be a promising source of myconutrients with functional and antioxidant properties.

Additionally, the correlation between the TAA and TPC together with individual phenolics and total phenolic acids were also established. Notably, 3,4-dihydroxybenzaldehyde (DHBA) showed strong significant (*p* < 0.0001) correlation with TAA (r = 0.89) and TPC (r = 0.61). Similarly, the salicylic acid (SalA) content in the extract was strongly correlated with the TAA (r = 0.89) and TPC (r = 0.88). The total hydroxybenzoic acids (ToHBen) was also significantly (*p* < 0.0001) correlated with TAA (r = 0.83) and TPC (0.54), respectively. That may explain the contributions of hydroxybenzoic acids to the total antioxidative capacity of the extract obtained using this solvent extraction system. On the contrary, the ferulic acid (FA) content in the extract was moderately, but significantly (*p* = 0.0001) correlated with TPC (r = 0.74) as was caffeic acid (CA) with TAA (r = 0.70). This trend in moderate association was consistent between the other phenolics and the TAA or TPC evaluated (Figure 3).

## 4. Conclusions

The SWP method of extraction using hot water with different pH values has been very useful in obtaining high-quality food-grade and green extracts from Chaga enriched with myconutrients. These extracts represent a good source of myconutrients such as phenolics, fat- and water-soluble vitamins, and minerals. The highest TPC and TAA levels were found in extracts using pH 11.5 and a temperature of 100 °C. This was the case also for most of the individual phenolics detected in Chaga. Altogether, 24 phenolic compounds were detected in the extract where 3,4-dihydroxybenzaldehyde (104.851 µg/mL) was found as the major phenolic, which was highly correlated with the TAA and TPC. Vitamins are first analyzed in Chaga, being the Vitamin E and K with the highest amounts found. Furthermore, Other phenolics like 4-methoxycinnamic acid, isorhamnetin, protocatechuic acid 4-glycoside, among others are described for the first time in Chaga using the modified Swiss water method evaluated in this study.

The Chaga extracts contained elevated TPC content and antioxidant properties at higher pH and temperature under pressurized extraction conditions. Treatment using pH 11.5 at 100 °C was useful for recovering the phenolics (total hydroxycinnamic acids, hydroxybenzoic acids, and total other phenolics) and total antioxidants. The antioxidant minerals were also best extracted at pH 11.5 at 100 °C, while the macro and trace minerals extraction, were more efficient using either pH 2.5 or 9.5 at 100 °C. This work demonstrates that water as pressurized fluid at high pH and temperature were very effective in extracting myconutrients from Chaga matrices and may also be useful for the recovery of bioactive nutrients from different functional food matrices. This work presents a modified Swiss water method at high pH (11.5) and temperature (100 °C) under pressurized conditions as a suitable green or food grade solvent for the extraction of antioxidant myconutrients (i.e., phenolics, Zn, K, Mn, and vitamin E) from Chaga matrices. These results suggest Chaga extracts obtained using the SWP could be promising sources of myconutrients or functional components that may have applications in producing nutraceuticals (i.e., Chaga capsules), or functional foods abundant with the antioxidants.

## Figures and Tables

**Figure 1 antioxidants-10-01322-f001:**
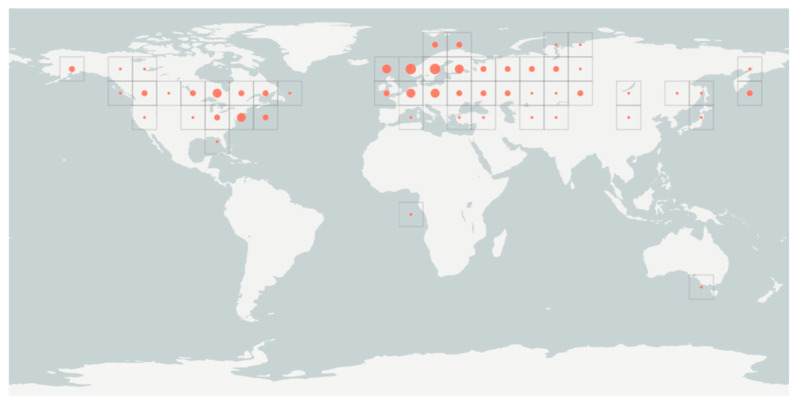
Global Geographical distribution of *Inonotus obliquus* in the circumboreal regions.

**Figure 2 antioxidants-10-01322-f002:**
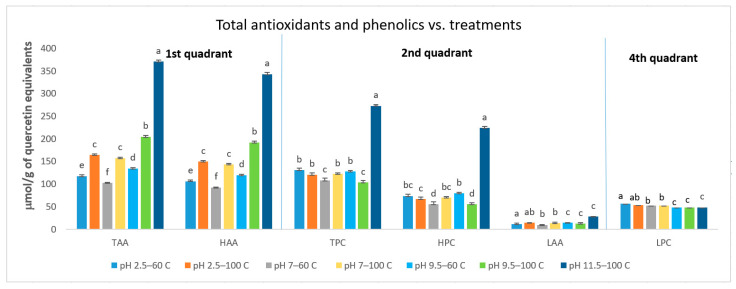
Bar charts showing the difference in phenolics and antioxidants following ANOVA for the components segregated per quadrant in the PCA biplot. Hydrophilic antioxidant activity (HAA), lipophilic antioxidant activity (LAA), total antioxidant (TAA), hydrophilic phenolic content (HPC), lipophilic phenolic content (LPC), and total phenolic content (TPC). The antioxidant activities are reported as μmol/g of Trolox equivalents. The phenolics content is represented as μmol/g of quercetin equivalents. Values accompanied by the same letters are not significantly different (*p* < 0.05), Fisher’s LSD test.

**Figure 3 antioxidants-10-01322-f003:**
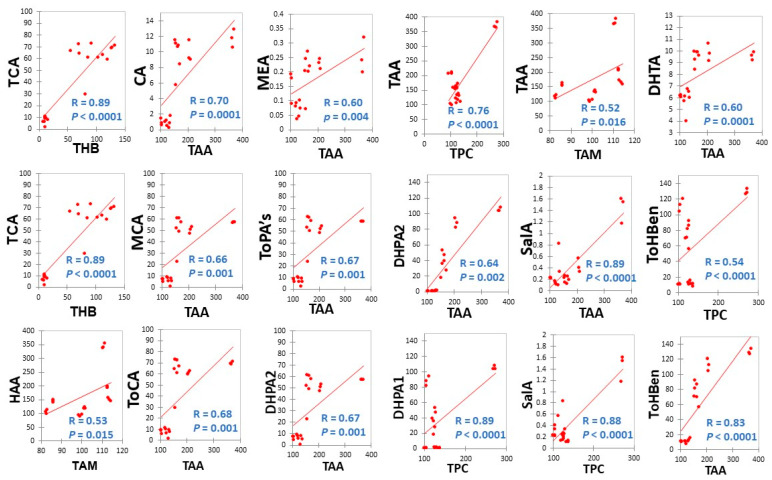
Correlation between TAA and CA, TAA and MEA, TAA and TPC, TAA and TAM, DHTA and TAA, MCA and TAA, ToPA’s and TAA, DHPA2 and TAA, SalA and TA, ToHBen and TPC, ToCA and TAA, DHPA2 and TAA, DHPA1 and TPC, SalA and TPC, ToHBen and TAA content. HAA = hydrophilic antioxidant activity, LAA = lipophilic antioxidant activity, TAA = total antioxidant activity, HPC = hydrophilic phenolic content, LPC = lipophilic phenolic content, TPC = total phenolic content. TAM = total antioxidant minerals, ToHBen = total hydroxybenzoic acids, ToPA’s = total other phenolic acids. ToCA = total cinnamic acids. DHPA1 = 3,4-dihydroxybenzaldehyde, DHPA2 = 3,4-dihydroxybenzalacetone, SalA = Salicylic acid, DHTA = dihydroxyterephtalic acid, MEA = methyl-gallic acid, CA = caffeic acid. Antioxidant activity was expressed as μmol Trolox equivalent/g extract of Chaga and phenolic content was expressed as μmol quercetin equivalent/g extract of Chaga. R-values represent Pearson correlation coefficients, *p* = 0.05, *n* = 4 per replicate of each sample at various pH values and temperatures.

**Figure 4 antioxidants-10-01322-f004:**
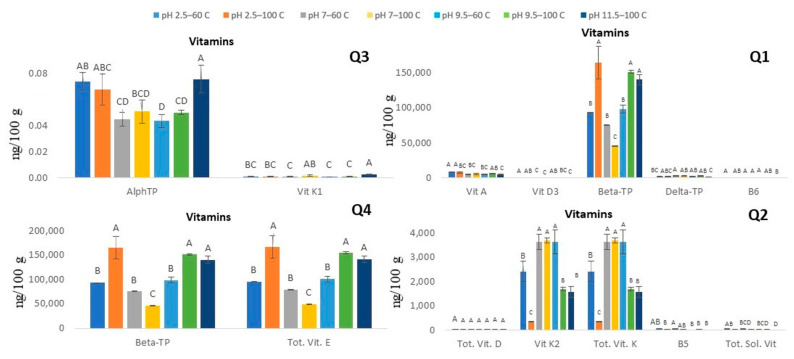
Vitamin contents in *Inonotus obliquus* samples extracted under different combinations of pH (2.5, 7, 9.5, 11.5) and temperatures (60 °C and 100 °C). Bars represent means of four replicates ± standard errors. Values accompanied by the same letters are not significantly different (*p* < 0.05), Fisher’s LSD test.

**Figure 5 antioxidants-10-01322-f005:**
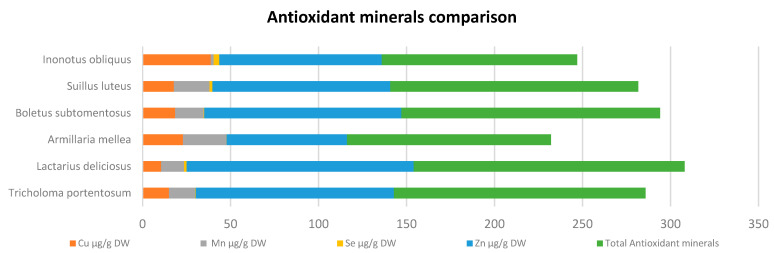
Antioxidant minerals comparison from different species [44].

**Table 1 antioxidants-10-01322-t001:** Mineral content in Chaga (*Inonotus Obliquus*) extracts as a function of solvent pH and temperature under pressurized conditions.

pH Treatment and Temperature **								
Classification of Minerals	pH 2.5 at 60 °C	pH 7 at 60 °C	pH 9.5 at 60 °C	pH 2.5 at 100 °C	pH 7 at 100 °C	pH 9.5 at 100 °C	pH 11.5 at 100 °C	RDAs *
**Antioxidant minerals**								
**^55^Mn**	9.85 ± 0.04 ^a^	0.53 ± ND ^b^	0.38 ± 0.01 ^c^	0.34 ± 0.01 ^d^	0.28 ± 0.14 ^e^	0.11 ± 0.08 ^f^	ND ***	2 (mg)
**^63^Cu**	2.10 ± 0.15 ^f^	3.84 ± 0.25 ^d^	4.24 ± 0.08 ^c^	4.27 ± 0.19 ^c^	3.45 ± 0.31 ^e^	4.38 ± 0.78 ^b^	4.82 ± 0.04 ^a^	1 (mg)
**^64^Zn**	67.37 ± 1.29 ^g^	91.86 ± 4.90 ^e^	93.20 ± 2.53 ^d^	105.87 ± 3.05 ^a^	79.75 ± 0.42 ^f^	104.75 ± 0.37 ^b^	103.05 ± 2.84 ^c^	10 (mg)
**^80^Se**	3.46 ± 0.07 ^ab^	3.05 ± 1.76 ^c^	3.63 ± 0.42 ^a^	3.37 ± 0.93 ^b^	2.42 ± 0.18 ^d^	3.51 ± 0.46 ^ab^	3.04 ± 0.11 ^c^	55 (µg)
**Macrominerals**								
**^23^Na**	1439.35 ± 5.91 ^c^	1406.28 ± 5.16 ^d^	1414.55 ± 11.03 ^cd^	1612.63 ± 1.21 ^a^	1228.21 ± 0.68 ^e^	1583.92 ± 2.11 ^ab^	1580.22 ± 23.89 ^b^	800 (mg)
**^24^Mg**	787.45 ± 2.15 ^d^	1028.89 ± 1.58 ^b^	1036.58 ± 1.35^b^	1155.43 ± 10.34 ^a^	891.44 ± 1.24 ^c^	1159.06 ± 7.20 ^a^	1161.53 ± 6.87 ^a^	375 (mg)
**^31^P**	896.61± 0.60 ^a^	339.63 ± ND*** ^c^	335.85 ± 15.82 ^c^	370.48 ± 2.40 ^b^	290.58 ± 0.46 ^d^	378.80 ± 9.31 ^b^	335.65 ± 3.00 ^c^	700 (mg)
**^39^K**	2004.07 ± 19.96 ^a^	940.47 ± 4.41 ^e^	1026.71 ± 9.75 ^d^	1853.77 ± 27.08 ^b^	902.02 ± 7.62 ^e^	1211.62 ± 2.28 ^c^	899.74 ± 4.81 ^e^	2000 (mg)
**^44^Ca**	480.10 ± 7.83 ^d^	660.32 ± 3.67 ^b^	671.57 ± 3.90 ^b^	745.19 ± 6.58 ^a^	577.58 ± 5.58 ^c^	756.34 ± 11.04 ^a^	750.47 ± 3.73 ^a^	800 (mg)
**Trace Minerals**								
**^11^B**	64.25 ± 17.44 ^a^	11.25 ± 2.57 ^c^	8.97 ± 3.86 ^d^	12.81 ± 5.68 ^bc^	10.21 ± 0.36 ^d^	15.26 ± 11.78 ^b^	13.84 ± ND ^b^	
**^52^Cr**	3.94 ± 0.49	ND	ND	ND	ND	ND	ND	40 (µg)
***Toxic Minerals***								
**^75^As**	1.63 ± 0.04 ^a^	0.94 ± 0.01 ^d^	0.97 ± 0.02 ^c^	1.03 ± 0.01 ^b^	0.81 ± 0.03 ^e^	1.04 ± 0.09 ^b^	1.04 ± 0.01 ^b^	

* RDA’s recommended daily allowances. Values were obtained from “Commission Regulation (EC) No 629/2008”. ** Values represent means of three replicates ± standard errors. For each column, values accompanied by the same letters are not significantly different (*p* < 0.05), Fisher’s LSD test. *** ND = not detected. Macro-mineral chlorine; trace minerals fluoride, uodine, and molybdenum; and toxic mineral. The Chaga extracts are at various pH values (2.5, 7, 9.5, and 11.5) and at temperatures 60 °C and 100 °C. Fe, Co, Cd, and Pb were not detected at any treatment. Highest values of Mn, P, K, B, Cr, and as were detected in pH 2.5 at 60 °C.

**Table 2 antioxidants-10-01322-t002:** Phenolics content (μg·g^−1^ dry weight) in *Inonotus obliquus* extracts obtained under different pH and temperature conditions.

Compounds	Formula	MW (g/mol)	*m/z* [M − H]^−^	RT (min)	Amount Recovered (µg/mL PA eq.)	Structure
**Hydroxybenzoic acids**						
Gallic acid_1Gallic acid_2Gallic acid_3	C_7_H_6_O_5_	170.12	169.0142	11.1213.3514.74	0.260.200.11	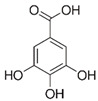
Protocatechuic acid	C_7_H_6_O_4_	154.12	153.0193	16.90	32.21	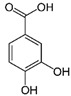
Salicylic acid	C_7_H_6_O_3_	138.122	137.0244	29.71	1.44	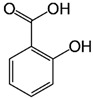
Vanillic acid_1Vanillic acid_2Vanillic acid_3Vanillic acid_4	C_8_H_8_O_4_	168.148	167.0350	19.48	0.470.450.330.18	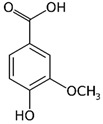
2,3-DihydroxybenzaldehydeDHBA1	C_7_H_6_O_3_	138.12	137.0244	18.28	104.85	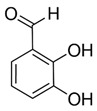
2,5-Dihydroxyterephthalic acid (DHTA)	C_8_H_6_O_6_	198.13	197.0092	16.00	9.83	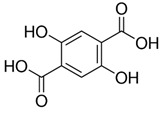
**Hydroxycinnamic** **acids**						
o-Coumaric acidp-Coumaric acid	C_9_H_8_O_3_	164.16	163.0401	18.1122.91	0.220.24	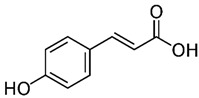
Caffeic acid	C_9_H_8_O_4_	180.16	179.0349	19.99	11.72	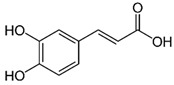
4-methoxycinnamic acid	C_10_H_10_O_3_	178.187	177.0557	21.11	0.57	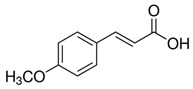
Hispidin	C_13_H_10_O_5_	246.22	245.0455	18.81	0.45	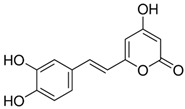
Ferrulic acid (trans)Ferrulic acid_1 (cis)	C_10_H_10_O_4_	194.18	193.0506	21.9017.10	0.400.18	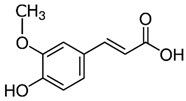
**Flavonoid derivatives**						
Isorhamnetin	C_16_H_12_O_7_	316.26	315.0510	23.34	0.47	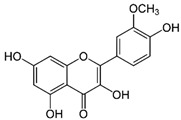
Myricetin_1Myricetin_2	C_15_H_10_O_8_	318.24	317.0303	22.4130.10	0.301.32	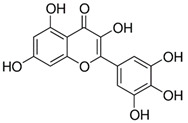
Quercetin	C_15_H_10_O_7_	302.236	301.0354	30.27	0.54	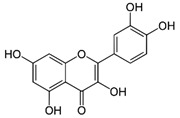
**Other phenolic acids**						
Syringic acid	C_9_H_10_O_5_	198.174	197.0455	17.30	1.14	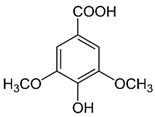
Ellagic acid	C_14_H_6_O_8_	302.197	300.9989	22.61	0.25	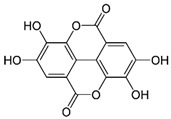
Hispolon	C_12_H_12_O_4_	220.224	219.0663	18.72	0.03	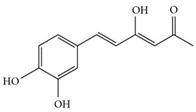
3,4-Dihydroxybenzalacetone(DHBA2)	C_10_H_10_O_3_	178.187	177.0557	19.03	57.17	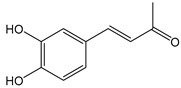
3-O-Methylellagic acid	C_15_H_8_O_8_	316.22	315.0146	25.55	0.25	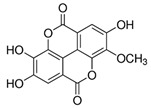
Total cinnamates		69.65	
Total Benzoate	129.20
Total flavonoids	1.80
Total other phenolics	58.24

## Data Availability

Data is contained within the article and Appendix A.

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
