# Peer review of "Effects of pH and Temperature on Water under Pressurized Conditions in the Extraction of Nutraceuticals from Chaga (Inonotus obliquus) Mushroom"

_antioxidants, 2021, doi:10.3390/antiox10081322_

Round 1
Reviewer 1 Report
Dear Authors,
The manuscript 1310256 “The influence of pH and temperature on water as a pressurized fluid in the extraction of bioactive myconutrients from chaga mushroom (Inonotus obliquus)” presents interesting data on the application of a modified Swiss water process (SWP) method to obtain extracts from Chaga under different pH and temperatures. Emphasis was given to total antioxidants activity, phenolics, vitamins, and minerals contents in Chaga-SWP-extracts.
If I correctly read the paper, Inonotus obliquus (Fr.) Pilát basidiomata were powdered and extracted using an Accelerated Solvent Extractor following a modified Swiss water process (SWP) method. Ionized water at different pH values (2.5, 7.0, 9.5 and 11.5) was used as extraction solvent. Two different temperatures (60 and 100⁰C) were tested. The lyophilized extracts were characterized as total antioxidant activity (TAA) and total phenolic content (TPC). Vitamins (A, B1, B2, B3, B5, B6, B9, B12, E, D3, K1, K2) and Phenolic compounds () were also quantified in the extracts by Ultra-High-pressure Liquid Chromatography-Mass Spectrometry. Minerals were also analysed and quantified by inductively coupled plasma-mass spectrometry (ICP-MS) on Chaga Mushroom extracts digested with 70% v/v nitric acid using a microwave digestion system. Twenty phenolic compounds were evaluated in the extracts and the highest TPC and TAA were observed at pH11.5 and 100°C. The most abundant phenolic compounds included protocatechuic acid 4-glucoside, syringic acid, and myricetin. Extractions at pH 2.5 and 100°C recovered the highest quantities of B vitamins, whereas treatments with ionized water at pH 2.5 at 60 and 100°C well recovered total fat-soluble vitamins. Vitamin E ranged between 50 to 175 μg/100g of dry Chaga basidiomata. Antioxidant minerals ranged from 85.94 μg/g (pH7 at 100°C) to 113.86 μg/g of dry basidiomata (pH2.5 at 100°C). The treatment at pH 11.5 and 100°C was the most useful for recovering phenolics and antioxidants from Chaga. SWP is proposed as a solvent system for the extraction of myco-nutrients with applications as a new and green food-grade approach to extract nutrients from food-matrices.
My general impression of this study is positive, but its presentation in the form of a manuscript requires adjustments.
Arrange the manuscript (reference indications) following the journal's instructions (https://www.mdpi.com/journal/antioxidants/instructions).
Use the International System of Units.
Lines: 293, 301, 338, 346, 366, 373, 426, 486, 551, 7, Uniform Chaga or chaga
The following suggestions could improve the manuscript.
I suggest the title:
Effects of pH and temperature on water under pressurized conditions in the extraction of nutraceuticals from chaga (Inonotus obliquus) mushroom
Abstract
Reduce the general information on mushroom importance. Avoid unused abbreviation.
Insert information on TAA and TPC. Explain the correlations between subclasses of bioactive myconutrients (phenolics, vitamins, and minerals) and their contributions to TAA and TPC.
Keywords
I suggest the following:
Accelerated Solvent Extraction (ASE), Functional foods; Swiss water process (SWP), Green extraction; Total antioxidant activity; Total phenolic content; B-complex Vitamins; Fat-Soluble Vitamins
Introduction
Use the correct Author’s name for Chaga: Inonotus obliquus (Fr.) Pilát (see: http://www.indexfungorum.org/names/Names.asp). In non-taxonomic paper is facultative indicate the Authors associated to Latin names of plants and fungi. Indicate the correct abbreviation or delete all the Authors name.
A brief description of Inonotus obliquus morphology and cultivation could help the reader to better understand the work.
Line 51: use “sporocarp” instead of “fruiting bodies”.
Line 58: delete “a fungus and”.
Figure 1 legend: Line 72: Use “Inonotus obliquus” instead of the local name “Chaga mushroom”. In fact, Chaga is the name transliterated from the Russian common name of the fungus. It is also known as the clinker polypore, "clinker", cinder conk, birch canker polypore, and sterile conk trunk rot of birch.
Line 74: Is the capital letter necessary for “Macrominerals”?
Line 78: delete “significant quantities of”.
Line 94: Insert appropriate reference/s.
Materials and Methods
Line 117: Use “Standard water” instead of “Water”
Line 135: Use “Chaga sporocarp” instead of “Chaga”.
Line 173: What does "ug" mean?
Line 220: Is the capital letter necessary for “Formic”?
Line 261: delete “One-way”.
In the “2.9. Statistical Analysis” describe how the principal component analysis (PCA) procedure was conducted. PCA was abundantly commented in the results section.
Results and discussion
This section is particularly difficult to read.
One suggestion might be to reduce the PCA part and comment directly on the data with the LSD test.
Standardize the extraction conditions using, for example, the association pH 2.5 and 60°C.
I suggest the following sub-sections:
3.1. Total Polyphenols Content (TPC) and Total Antioxidant Activity (TAA)
3.2. Chaga extracts composition
3.2.1. Minerals
3.2.2. Vitamins (water- and fat-soluble)
3.2.3. Phenolics
3.2.4. Correlations (I suggest avoiding this section and insert the correlation under the previous sections)
Line 314: What does " vital" mean?
Line 315: Is the capital letter necessary for “Mushroom.”?
Line 336: Is Figure 2A necessary?
Insert a figure legend.
Lines 284-285: delete
Line 386: Is Figure 3 necessary?
If figure 3 and Table 1 show the same data, I suggest to delete the figure.
Table 1
What do the bold values indicate?
Put "Antioxidants mineral" on the same line
I suggest the following legend and table notes:
Table 1. Mineral content (μg g-1 dry weight) in Inonotus obliquus extracts obtained under different pH and temperature conditions**
Use “ND***” instead of “0.00”
table notes:
*RDA’s recommended daily allowances. Values were obtained from “Commission Regulation (EC) No 629/2008”.
** Values represent means of three replicates ± standard errors. For each column, values accompanied by the same letters are not significantly different (P < 0.05), Fisher’s LSD test.
*** ND = not detected.
Line 465: Is Figure 4a necessary?
Figure 4
I suggest the following legend:
Figure 4. Vitamin contents in Inonotus obliquus samples extracted under different combinations of pH (2.5, 7, 9.5, 11.5) and temperatures (60°C and 100°C). Bars represent means of four replicates ± standard errors. Values accompanied by the same letters are not significantly different (P < 0.05), Fisher’s LSD test.
Line 505: Is Figure 5a necessary?
If figure 5b and Table 2 show the same data, I suggest to delete the figure and reduce formulas size in the table.
Table 2
I suggest the following legend:
Table 2. Phenolics content (μg g-1 dry weight) in Inonotus obliquus extracts obtained under different pH and temperature conditions**
It is preferable to list the phenol molecules according to their chemical nature (Cinnamic acids derivatives, Benzoic acids derivative, Flavonoids and Other phenolics).
Figure 7 a: A table makes data more readable and comparable.
Delete Fig 7b and 7C
Figure 6
A table with Pearson correlation coefficient and its P value improves data readability
Conclusion
This section is an abstract. Rewrite based on all suggested revisions.
References
Arrange all the references following the journal's instructions.
Best regards
Author Response
Comments and Suggestions for Authors
Dear Authors,
The manuscript 1310256 “The influence of pH and temperature on water as a pressurized fluid in the extraction of bioactive myconutrients from chaga mushroom (Inonotus obliquus)” presents interesting data on the application of a modified Swiss water process (SWP) method to obtain extracts from Chaga under different pH and temperatures. Emphasis was given to total antioxidants activity, phenolics, vitamins, and minerals contents in Chaga-SWP-extracts.
If I correctly read the paper, Inonotus obliquus (Fr.) Pilát basidiomata were powdered and extracted using an Accelerated Solvent Extractor following a modified Swiss water process (SWP) method. Ionized water at different pH values (2.5, 7.0, 9.5 and 11.5) was used as extraction solvent. Two different temperatures (60 and 100⁰C) were tested. The lyophilized extracts were characterized as total antioxidant activity (TAA) and total phenolic content (TPC). Vitamins (A, B1, B2, B3, B5, B6, B9, B12, E, D3, K1, K2) and Phenolic compounds () were also quantified in the extracts by Ultra-High-pressure Liquid Chromatography-Mass Spectrometry. Minerals were also analysed and quantified by inductively coupled plasma-mass spectrometry (ICP-MS) on Chaga Mushroom extracts digested with 70% v/v nitric acid using a microwave digestion system. Twenty phenolic compounds were evaluated in the extracts and the highest TPC and TAA were observed at pH11.5 and 100°C. The most abundant phenolic compounds included protocatechuic acid 4-glucoside, syringic acid, and myricetin. Extractions at pH 2.5 and 100°C recovered the highest quantities of B vitamins, whereas treatments with ionized water at pH 2.5 at 60 and 100°C well recovered total fat-soluble vitamins. Vitamin E ranged between 50 to 175 μg/100g of dry Chaga basidiomata. Antioxidant minerals ranged from 85.94 μg/g (pH7 at 100°C) to 113.86 μg/g of dry basidiomata (pH2.5 at 100°C). The treatment at pH 11.5 and 100°C was the most useful for recovering phenolics and antioxidants from Chaga. SWP is proposed as a solvent system for the extraction of myco-nutrients with applications as a new and green food-grade approach to extract nutrients from food-matrices.
-My general impression of this study is positive, but its presentation in the form of a manuscript requires adjustments. Arrange the manuscript (reference indications) following the journal's instructions (https://www.mdpi.com/journal/antioxidants/instructions).
Response:
We would thank the reviewer for his/her comments. We have revised the manuscript references, accordingly.
-Use the International System of Units.
Response:
Done
-Lines: 293, 301, 338, 346, 366, 373, 426, 486, 551, 7, Uniform Chaga or chaga
Response:
Done
-I suggest the title:
Effects of pH and temperature on water under pressurized conditions in the extraction of nutraceuticals from chaga (Inonotus obliquus) mushroom
Response:
In agreement with the reviewer recommendation, we have used the proposed title as it better reflects the manuscript’s content.
Abstract
-Reduce the general information on mushroom importance. Avoid unused abbreviation.
Response:
Done
-Insert information on TAA and TPC. Explain the correlations between subclasses of bioactive myconutrients (phenolics, vitamins, and minerals) and their contributions to TAA and TPC.
Response:
In line with the reviewer’s recommendation, we have added a paragraph regarding the TAA and TPC
Keywords
-I suggest the following:
Accelerated Solvent Extraction (ASE), Functional foods; Swiss water process (SWP), Green extraction; Total antioxidant activity; Total phenolic content; B-complex Vitamins; Fat-Soluble Vitamins
Response:
We have used the suggested Keywords, accordingly.
Introduction
-Use the correct Author’s name for Chaga: Inonotus obliquus (Fr.) Pilát (see: http://www.indexfungorum.org/names/Names.asp). In non-taxonomic paper is facultative indicate the Authors associated to Latin names of plants and fungi. Indicate the correct abbreviation or delete all the Authors name.
Response:
Done, we deleted the author’s name.
-A brief description of Inonotus obliquus morphology and cultivation could help the reader to better understand the work.
Response:
Some morphological info. has been included into the text.
-Line 51: use “sporocarp” instead of “fruiting bodies”.
Response:
Done
-Line 58: delete “a fungus and”.
Response:
Done
-Figure 1 legend: Line 72: Use “Inonotus obliquus” instead of the local name “Chaga mushroom”. In fact, Chaga is the name transliterated from the Russian common name of the fungus. It is also known as the clinker polypore, "clinker", cinder conk, birch canker polypore, and sterile conk trunk rot of birch.
Response:
Done
-Line 74: Is the capital letter necessary for “Macrominerals”?
Response:
Not really, the capital letter was replaced with a small letter along the manuscript.
-Line 78: delete “significant quantities of”.
Response:
Done
-Line 94: Insert appropriate reference/s.
Response:
Done
Materials and Methods
-Line 117: Use “Standard water” instead of “Water”
Response:
Done
-Line 135: Use “Chaga sporocarp” instead of “Chaga”.
Response:
Done
-Line 173: What does "ug" mean?
Response:
It was correctly replaced with µg
-Line 220: Is the capital letter necessary for “Formic”?
Response:
Not really, we modified it accordingly.
-Line 261: delete “One-way”.
Response:
Done
-In the “2.9. Statistical Analysis” describe how the principal component analysis (PCA) procedure was conducted. PCA was abundantly commented in the results section.
Results and discussion
-One suggestion might be to reduce the PCA part and comment directly on the data with the LSD test.
Standardize the extraction conditions using, for example, the association pH 2.5 and 60°C.
Response:
In agreement with the reviewer recommendation, we have reduced the PCA commenting and put more focus on the quantitative results.
-I suggest the following sub-sections:
3.1. Total Polyphenols Content (TPC) and Total Antioxidant Activity (TAA)
3.2. Chaga extracts composition
3.2.1. Minerals
3.2.2. Vitamins (water- and fat-soluble)
3.2.3. Phenolics
3.2.4. Correlations (I suggest avoiding this section and insert the correlation under the previous sections)
Response:
In agreement with the reviewers’ suggestions, we have adopted all the changes requested.
-Line 314: What does " vital" mean?
Response:
Vital has been replaced with useful
-Line 315: Is the capital letter necessary for “Mushroom.”?
Response:
The capital letter was replaced with a small letter
-Line 336: Is Figure 2A necessary?
Response:
We have deleted Fig. 2A
-Insert a figure legend.
Response:
Fig. legend was added for Fig. 2.
-Lines 284-285: delete
Response:
Lines 284-285 were deleted from the text.
-Line 386: Is Figure 3 necessary?
Response:
We have deleted fig. 3, accordingly.
-If figure 3 and Table 1 show the same data, I suggest to delete the figure.
Response:
In agreement with the reviewer suggestions, we have deleted fig. 3.
Table 1
-What do the bold values indicate?
Response:
We have changed the bold values correctly.
Put "Antioxidants mineral" on the same line
Response:
Done
-I suggest the following legend and table notes:
Table 1. Mineral content (μg g-1 dry weight) in Inonotus obliquus extracts obtained under different pH and temperature conditions**
Response:
Done, it was added accordingly
-Use “ND***” instead of “0.00”
Response:
Done, it was added accordingly
-table notes:
*RDA’s recommended daily allowances. Values were obtained from “Commission Regulation (EC) No 629/2008”.
Response: Done, it was added accordingly
-** Values represent means of three replicates ± standard errors. For each column, values accompanied by the same letters are not significantly different (P < 0.05), Fisher’s LSD test.
Response: Done, it was added accordingly
-*** ND = not detected.
Response:
Done, it was added accordingly
-Line 465: Is Figure 4a necessary?
Response:
We have deleted the fig. 4a.
Figure 4
-I suggest the following legend:
Figure 4. Vitamin contents in Inonotus obliquus samples extracted under different combinations of pH (2.5, 7, 9.5, 11.5) and temperatures (60°C and 100°C). Bars represent means of four replicates ± standard errors. Values accompanied by the same letters are not significantly different (P < 0.05), Fisher’s LSD test.
Response:
We have used the legend suggested by the reviewer.
-Line 505: Is Figure 5a necessary?
Response:
Figure 5 was deleted.
-If figure 5b and Table 2 show the same data, I suggest to delete the figure and reduce formulas size in the table.
Response:
In agreement with the reviewer suggestion, we have reduced the formula size and delete the figure 5.
Table 2
-I suggest the following legend:
Table 2. Phenolics content (μg g-1 dry weight) in Inonotus obliquus extracts obtained under different pH and temperature conditions**
Response:
In line with the reviewer suggestion, we have replaced the legend.
-It is preferable to list the phenol molecules according to their chemical nature (Cinnamic acids derivatives, Benzoic acids derivative, Flavonoids and Other phenolics).
Response:
The phenolic compounds have been classified in Table 2 as per the reviewer recommendation.
-Figure 7 a: A table makes data more readable and comparable.
Response:
In agree with the reviewer, we have added a table to the supplementary material
-Delete Fig 7b and 7C
Response:
Done
Figure 6
-A table with Pearson correlation coefficient and its P value improves data readability
Response:
In line with the reviewer suggestion, tables for Pearson correlation coefficient and its P value were added as a supplementary material as Table S3.A and Table S3.B
Tables for
Conclusion
-This section is an abstract. Rewrite based on all suggested revisions.
Response:
In agreement with the reviewer recommendations, we have rewritten the conclusion
References
-Arrange all the references following the journal's instructions.
Response:
Done
Reviewer 2 Report
Review of the manuscript entitled „The influence of pH and temperature on water as a pressurized fluid in the extraction of bioactive myconutrients from chaga mushroom (Inonotus obliquus)”. The topic of inonotus obliquus (IO) is close to my heart and is extremely important in folk medicine. In folk medicine, IO is usually used in the form of tea - an aqueous solution; Therefore, the topic taken up by the authors is very important. The manuscript prepared by the Authors is very interesting, but minor corrections are required.
First of all literature and citation in the text needs to be improved. Please use Arabic ([5] or [6]) numerals, not Roman ([V] or [VI]) as it is now. It is now generally accepted style all over the world. The Roman style was good two thousand years ago, but it can be a problem for modern people.
Lines 80, 148, 276, 311, 352, 392, 420, 452, 502 reference lack of parenthesis.
Lines 90 and 94, if multiple methods are mentioned, references should be given in line 90 and 94
Lines 104, 348, 538, 545, 606, 609, 619 there is a double space, it should be corrected
Lines 136-138, this is the weakest part of the manuscript. Authors should know exactly where the sample is coming from, maybe it is from Europe, maybe Asia, maybe North America. Mentioned information must be given, at least from which area/region/country. The advantage is that samples are characterized very well by chromatography, but at least the region of sample origin must be indicated.
Line 335-337 (and all figures) - there is no description of the figures, each figure should be signed with a legend and well characterized, the axes in the charts are invisible, charts should generally be more refined, Authors should work on the professionalism of the figures
Table 1, we write the elements/minerals differently, e.g. “Mn55” where 55 is in subscript, and so for all
Table 2, in formula subscript must be included, something is wrong in the last lines of table 2
Line 563-564, incorrect citation style
Why fig 6 is after fig 7?
Fig. 6 legend sign “=” should be changed to “-“
Could the authors indicate a practical application of their method for the average user? If so, add it to the conclusion.
Author Response
Comments and Suggestions for Authors
-Review of the manuscript entitled „The influence of pH and temperature on water as a pressurized fluid in the extraction of bioactive myconutrients from chaga mushroom (Inonotus obliquus)”. The topic of inonotus obliquus (IO) is close to my heart and is extremely important in folk medicine. In folk medicine, IO is usually used in the form of tea - an aqueous solution; Therefore, the topic taken up by the authors is very important. The manuscript prepared by the Authors is very interesting, but minor corrections are required.
Response:
We would thank the reviewer for the kind words, comments, and recommendations raised.
-First of all literature and citation in the text needs to be improved. Please use Arabic ([5] or [6]) numerals, not Roman ([V] or [VI]) as it is now. It is now generally accepted style all over the world. The Roman style was good two thousand years ago, but it can be a problem for modern people.
Response:
In agreement with the reviewer recommendation, we have revised the manuscript accordingly and corrected to include the Arabic numbers.
-Lines 80, 148, 276, 311, 352, 392, 420, 452, 502 reference lack of parenthesis.
Response:
Parenthesis have been added to the references
-Lines 90 and 94, if multiple methods are mentioned, references should be given in line 90 and 94
Response:
In line with the reviewer recommendations, references were merged into the text.
-Lines 104, 348, 538, 545, 606, 609, 619 there is a double space, it should be corrected
Response:
Done
-Lines 136-138, this is the weakest part of the manuscript. Authors should know exactly where the sample is coming from, maybe it is from Europe, maybe Asia, maybe North America. Mentioned information must be given, at least from which area/region/country. The advantage is that samples are characterized very well by chromatography, but at least the region of sample origin must be indicated.
Response:
The Chaga samples were obtained from Canada, specifically from Newfoundland province. It was included in the text, accordingly.
-Line 335-337 (and all figures) - there is no description of the figures, each figure should be signed with a legend and well characterized, the axes in the charts are invisible, charts should generally be more refined, Authors should work on the professionalism of the figures
Response:
In agreement with the reviewer, we have revised the figures and added the legends for the figures.
-Table 1, we write the elements/minerals differently, e.g. “Mn55” where 55 is in subscript, and so for all
Response:
Done
-Table 2, in formula subscript must be included, something is wrong in the last lines of table 2
Response:
In agreement with the reviewer, we have converted the formula format numbers as subscript.
-Line 563-564, incorrect citation style
Response:
Corrected
-Why fig 6 is after fig 7?
Response:
The order has been corrected in the text and in the figure captions
-Fig. 6 (now fig. 4) legend sign “=” should be changed to “-“
Response:
Done
-Could the authors indicate a practical application of their method for the average user? If so, add it to the conclusion.
Response:
In fact, this method is newly proposed for such a purpose of extraction of active compounds.
Actually, now, we are unable to indicate a specific application, as this needs to be an optimized method for such a scale of extraction. We may propose Chaga capsules as a source of nutraceuticals.
Round 2
Reviewer 2 Report
Thanks to the Authors for making corrections but now two things are missing. Between line 343 and 344 figure description is missing (probably number two). Why figure not have a Q3?
Please remove the negative units (-10000) from figure 3. Vitamins cannot take negative values.
The citation style in figure 4 is incorrect (Mirończuk-Chodakowska, Socha, Zujko, Terlikowska,
Borawska, Witkowska, 2019). Should be in “[…]”.
Author Response
Thanks to the Authors for making corrections but now two things are missing. Between line 343 and 344 figure description is missing (probably number two). Why figure not have a Q3?
Response: We would thank the referee for the revision and the comments. Figure caption and description have been added into the text properly. Regarding the Q3, no compounds were shown up in this quartile.
Please remove the negative units (-10000) from figure 3. Vitamins cannot take negative values.
Response: In agreement with the referee's suggestion, negative values were deleted.
The citation style in figure 4 is incorrect (Mirończuk-Chodakowska, Socha, Zujko, Terlikowska,
Borawska, Witkowska, 2019). Should be in “[…]”.
Response: The reference style was modified accordingly.